# Real-world time-travel experiment shows ecosystem collapse due to anthropogenic climate change

Guandong Li [1] ✉, Torbjörn E. Törnqvist [1] & Sönke Dangendorf [2]

Predicting climate impacts is challenging and has to date relied on indirect methods, notably modeling. Here we examine coastal ecosystem change during 13 years of unusually rapid, albeit likely temporary, sea-level rise ($>10$ mm yr$^{-1}$) in the Gulf of Mexico. Such rates, which may become a persistent feature in the future due to anthropogenic climate change, drove rising water levels of similar magnitude in Louisiana's coastal wetlands. Measurements of surface-elevation change at 253 monitoring sites show that 87% of these sites are unable to keep up with rising water levels. We find no evidence for enhanced wetland elevation gain through ecogeomorphic feedbacks, where more frequent inundation would lead to enhanced biomass accumulation that could counterbalance rising water levels. We attribute this to the exceptionally rapid sea-level rise during this time period. Under the current climate trajectory (SSP2-4.5), drowning of ~75% of Louisiana's coastal wetlands is a plausible outcome by 2070.

Climate change increasingly affects numerous aspects of life on Earth, by means of – among others – heat stress, water scarcity, food security risks, infectious diseases, and threats to biodiversity and ecosystems[1]. While research over the past half-century has focused heavily on improving predictions about the climate system itself, the broad spectrum of climate impacts has motivated a wide range of studies that aim to understand how these impacts, such as those listed above, will play out in the future.

Sea-level rise and its threat to low-elevation coastal zones ranks among the most severe consequences of climate change due to its expected role in driving human migration[2], along with its detrimental impact on coastal ecosystems that rank among the most valuable on the planet[3]. While the magnitude and rate of future sea-level rise are not precisely known, it is one of the most predictable elements of the climate system and committed to rise for at least several centuries[4,5], owing to the slow response time of the cryosphere-ocean system.

Assessments of the future impact of sea-level rise on coastal ecosystems have significantly increased in number over the past decade[6,7] and include a variety of model studies[8–11] as well as the examination of past analogs of future sea-level rise, notably from the last deglaciation[12]. While these approaches have offered valuable insights, they also come with inherent limitations. Analogs from the geologic past concern pristine coastal settings that were likely much more resilient to environmental change than the heavily human-perturbed coastal zones that dominate the Earth's shorelines today. In other words, studies of such analogs tend to result in overly optimistic estimates of their vulnerability[13]. Meanwhile, models must be calibrated and validated against known conditions, but projections for the future involve conditions for which observations generally do not exist. While model validation by means of the geologic record can increase confidence (as is commonly done with climate models) this also concerns conditions with little if any anthropogenic influence. Therefore, even if the trajectory of climate change were to be precisely known, considerable uncertainties remain about its impacts on ecosystems that have suffered degradation due to human activity.

Based on the estimated monetary value of coastal ecosystems[3], the ~15,000 km$^2$ of coastal wetlands in Louisiana (Fig. 1A) provide goods and services as high as $300B per annum. Because this region has seen the highest rates of wetland loss in the world over the past century[14,15], it has long been recognized[16,17] that this is a problem of

[1]Department of Earth and Environmental Sciences, Tulane University, 6823 St. Charles Avenue, New Orleans, LA 70118-5698, USA. [2]Department of River-Coastal Science and Engineering, Tulane University, 6823 St. Charles Avenue, New Orleans, LA 70118-5698, USA. ✉e-mail: gli4@tulane.edu

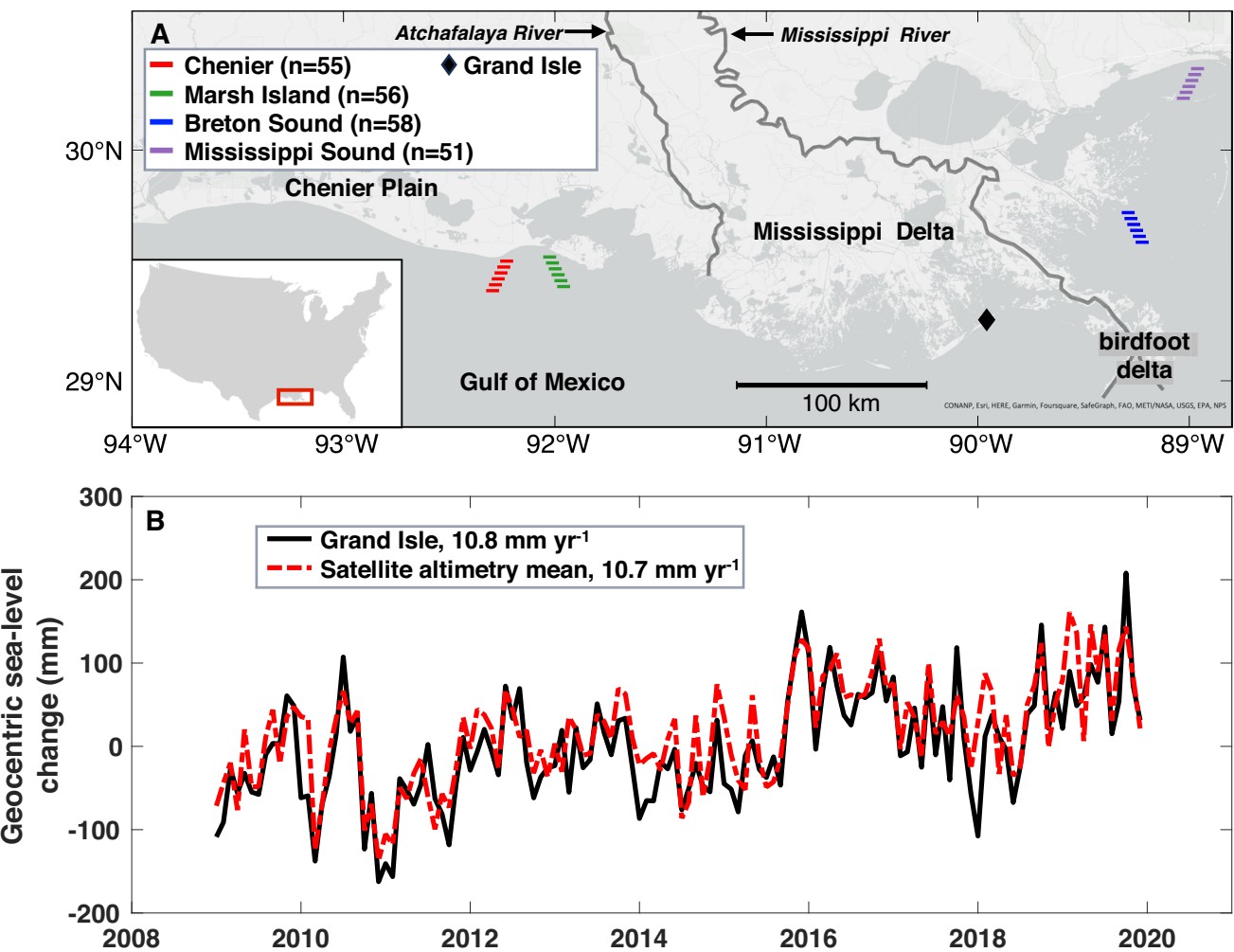

**Fig. 1 | Study area and location of satellite altimetry virtual stations. A** Map with the four satellite altimetry virtual stations (the number of observation points within each virtual station is shown in the legend) plus the Grand Isle tide gauge. **B** Mean geocentric sea-level change from all four satellite altimetry virtual stations and the Grand Isle tide gauge from 2009 to 2019.

utmost scientific and societal importance. While the causes of wetland loss are complex[18], fluctuations in the rate of accelerating sea-level rise have been highlighted as a contributing factor[19]. Previous work in this region that examined a landscape heavily impacted by river embankment, canal dredging, and other human actions reported that roughly half of the sites that were studied cannot keep pace with rising sea level today[20]. However, due to the scarcity of studies on regional geocentric sea-level (GSL) rise at the time, a highly conservative rate of 2 mm yr$^{-1}$ was adopted, precluding a full understanding of wetland response to rates of sea-level rise expected toward the end of this century. In addition, one uniform value for GSL rise was used[20], even though it is conceivable that local water-level changes at individual wetland sites may vary.

Here we investigate the impact of the exceptionally high rates of GSL rise (>10 mm yr$^{-1}$) that have affected the US Gulf Coast since about 2010[21,22]. As shown by these recent studies, this is likely a transient phenomenon caused by multidecadal, cyclic sea-level variability associated with ocean dynamics superimposed on the climate-driven acceleration that is seen worldwide. Thus, it is unlikely that these high rates will persist in the next few decades[21]. Nevertheless, this period can serve as an analog for the persistent rates of climate-driven sea-level rise that are expected later during this century and beyond. We therefore view this as a full-scale experiment of the response of an iconic ecosystem to future climate forcing. We examine the impact of 13 years of rapid GSL rise on coastal wetlands

that have additionally been experiencing subsidence rates averaging ~10 mm yr$^{-1}$ [20], along with human-caused degradation. This approach enables us to comprehensively address the question of whether coastal wetlands can adjust vertically by increased plant productivity and sedimentation[23], a feedback mechanism often cited to enhance wetland resilience[24]. We believe that the analysis presented herein opens a window to the future in a way that has to date not been possible. A rare exception is the work conducted on the Solomon Islands in the western Pacific where anomalously high rates of GSL rise (7–10 mm yr$^{-1}$ between 1994 and 2014) have resulted in a rapid decline or complete disappearance of reef islands[25]. This highlights the vulnerability of coastal communities to conditions expected by the end of this century, even in settings with limited anthropogenic disturbance.

We first examine how the accelerated GSL rise in the Gulf of Mexico has propagated into Louisiana's coastal wetlands (marshes and swamps) before analyzing the relationship between water-level rise and surface-elevation change at the local scale. The first objective is to answer the simple question of whether these wetlands have been able to keep up with the recent GSL rise. Among others, this allows us to test the hypothesis that coastal wetlands can adjust vertically to accelerated GSL rise by means of ecogeomorphic feedbacks. Finally, we address the question of when in the future—depending on the climate scenario—these conditions are likely to become persistent along the Gulf Coast.

## Results and discussion

### Propagation of sea-level rise into coastal wetlands

The rapid acceleration of GSL rise along the Gulf Coast commenced in 2010[21], with a rate at the Grand Isle tide gauge between 2009 and 2019 of 10.8 mm yr[-1] (Fig. 1B). To validate this rate, we examine the coastal GSL change from reprocessed satellite altimetry data[26] covering the period January 2009–December 2019 (see "Methods"). Four satellite altimetry virtual stations near the Louisiana coast provide monthly GSL change at points as close as 2 km from the shoreline to 20 km offshore, and more than 50 points are included within each of these 20 km coastal bands (Fig. 1A and Supplementary Fig. 1).

The GSL changes at the Grand Isle tide gauge and mean GSL changes from all four virtual stations (Fig. 1B) are highly correlated (r = 0.8, p < 0.001). Comparison of GSL change at Grand Isle and individual point measurements within the four virtual stations suggest lower correlation coefficients and higher standard deviations near the shoreline than farther offshore (Supplementary Fig. 2), likely due to complex coastal geomorphology, shallow water depth, and land contamination of the satellite altimetry signal. Rates of GSL change for each individual point (n = 220) (Supplementary Fig. 1) fluctuate around 10 mm yr[-1], with a mean rate of 10.7 mm yr[-1] (Fig. 1B), closely mimicking the 10.8 mm yr[-1] GSL rise at the Grand Isle tide gauge (see Methods).

We next assess relative water-level (RWL) change within the Louisiana coastal wetlands by analyzing data from water-level gauges at 325 monitoring sites from 2009 to 2021 (see "Methods"). The median rate of RWL change is 15.7 mm yr[-1] (mean: 16.7 ± 7.0 mm yr[-1]) with more than 97% of these sites showing statistically significant RWL rise (p < 0.001; Fig. 2A). During the same period, the GSL rise at the

Grand Isle tide gauge is 10.5 mm yr[-1]. This difference is because the water-level gauges also capture a significant portion of subsidence. For example, the high rates of RWL rise in the birdfoot delta (~30 mm yr[-1]) are very likely due to compaction within the ~100-m-thick Holocene sediment column in this area[27] (Fig. 2A). Rapid RWL rise is also found in an impounded portion of the Chenier Plain which is heavily impacted by manmade water control structures[28]. This is the area where monitoring sites exhibit lower correlation coefficients between RWL change and GSL change at the Grand Isle tide gauge (Fig. 2B). In the Mississippi Delta, a higher correlation is observed closest to the coast, suggesting a decreasing ocean influence farther inland. At the birdfoot delta, slightly lowered correlation coefficients may be related to elevated water levels during prolonged floods[29] (Fig. 2B). In summary, RWL changes across the Louisiana coast show that 97.2% of sites exhibit a statistically significant correlation (p < 0.001) with GSL change at the Grand Isle tide gauge, with a median correlation coefficient of 0.74 (Fig. 2B). Hence, GSL rise is the main factor driving RWL rise in coastal Louisiana, except in the highly impounded Chenier Plain[28] and near river mouths.

### Wetland response to accelerated sea-level rise

Commonly, surface-elevation change (SEC) as measured by means of the rod surface-elevation table (RSET) is compared to the relative sea-level (RSL, i.e., GSL plus vertical land motion) trend from the nearest tide gauge to evaluate wetland vulnerability[9,23,24,30–32]. Rather than relying on a limited number of tide gauges, here we use the RWL record from the water-level gauge associated with each monitoring site (n = 253), on average located only 93 m from the associated RSET

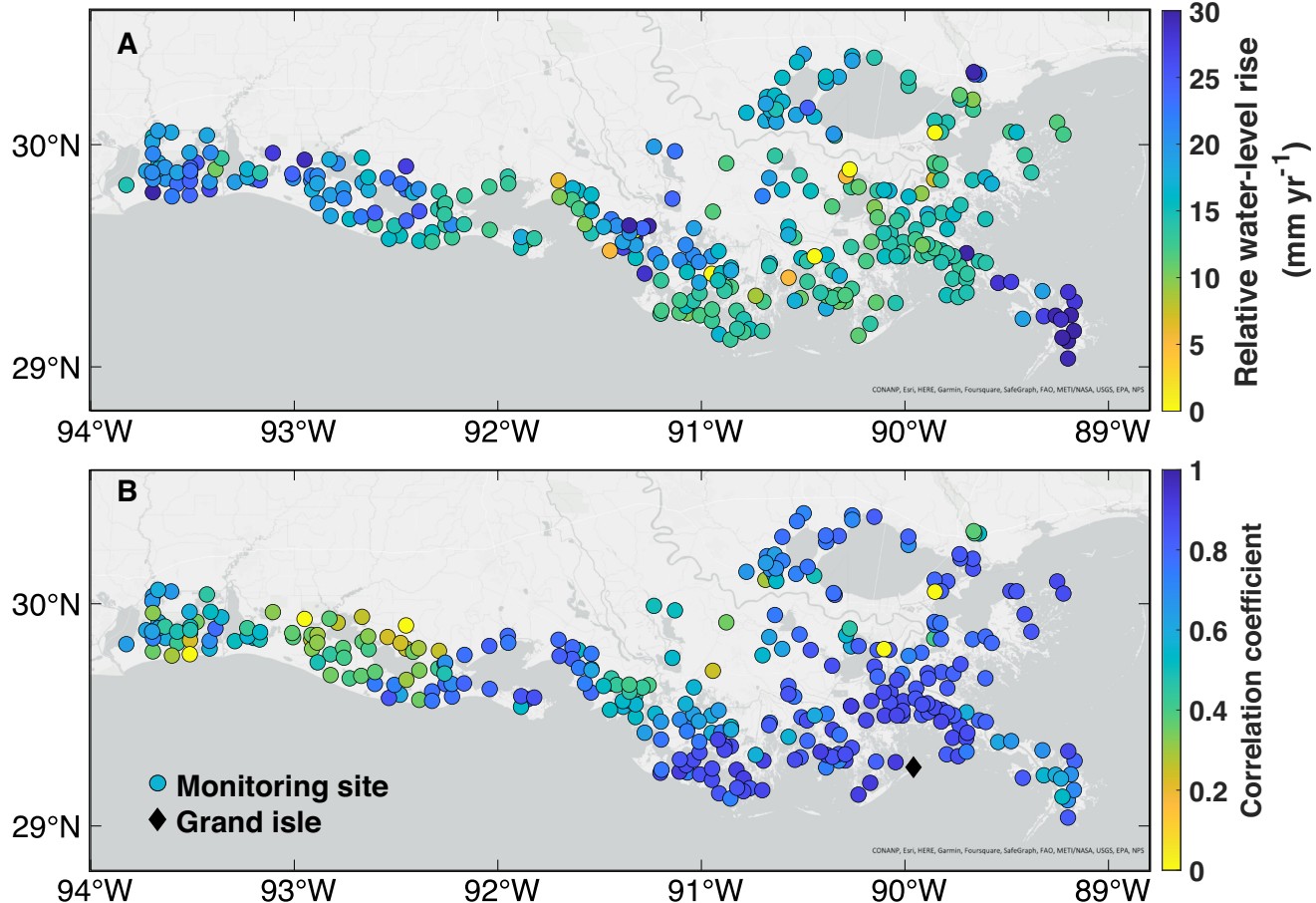

**Fig. 2 | Relative water-level (RWL) trends at monitoring sites and correlation with geocentric sea-level (GSL) change. A** Annual RWL trend from 2009 to 2021 (four sites with negative trends are plotted in bright yellow and nine sites with rates >30 mm yr[-1] are plotted in dark blue to avoid distortion of the scale). **B** Correlation coefficient between detrended RWL change at each monitoring site (n = 325) and detrended GSL change from the Grand Isle tide gauge between 2009 and 2021.

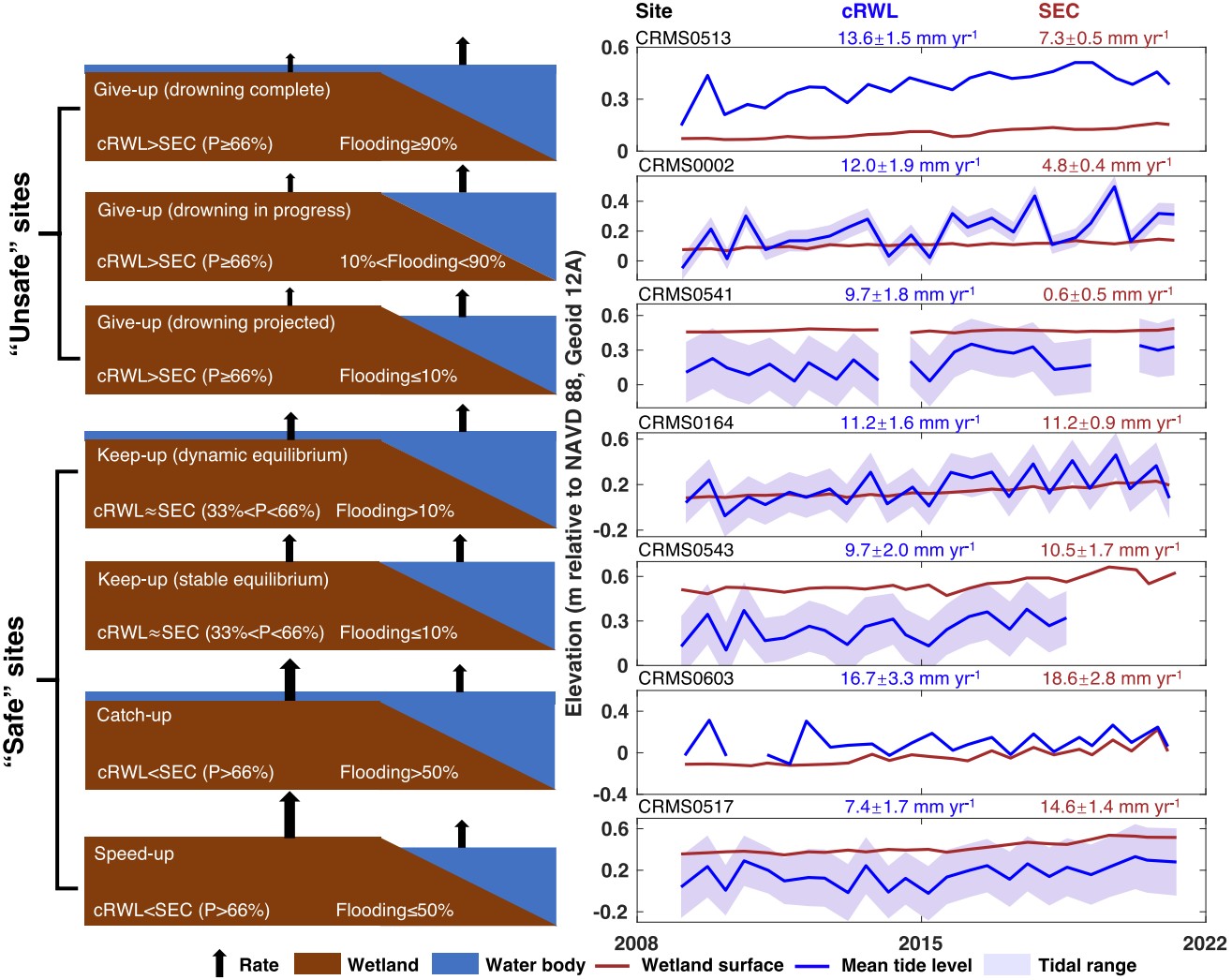

**Fig. 3 | Possible wetland responses by means of rates of surface-elevation change (SEC) to corrected relative water-level (cRWL) change.** P represents the probability of the specified cRWL-SEC relationship. Characteristic examples from individual monitoring sites are shown on the right with the rate (and one standard deviation) of cRWL change and SEC (see Methods for further details). The location of the example monitoring sites are shown in Supplementary Fig. 6.

(Supplementary Fig. 3). Our SEC time series average 12 years in length, i.e., well over the five years previously found to be the minimum to obtain robust SEC data[20]. In addition, we correct the RWL data by removing the vertical land motion that occurs between the base of the water-level gauge and the RSET (see "Methods"). Then, we compare the rates of wetland SEC and cRWL (corrected relative water-level) change, along with the elevation difference between the wetland surface and the adjacent water surface at each monitoring site, and we consider uncertainties through Monte Carlo simulation (see "Methods"). We then identify different wetland responses by adopting terminology developed to examine coral-reef response to sea-level change[33,34]. We classify each monitoring site into "give-up," "keep-up," or "catch-up," (including a few subdivisions) and add a "speed-up" category (Fig. 3).

With this classification scheme, the full spectrum of possible wetland responses to cRWL change is captured (Fig. 3). The give-up categories reveal that the wetland is unable to keep pace with cRWL rise (i.e., there is a surface-elevation deficit), with the elevation comparison between the wetland and water surface determining whether drowning currently occurs. On the other hand, the keep-up categories suggest that the wetland keeps pace with the cRWL rise, at least for the time interval under consideration. The catch-up category demonstrates that the wetland is gaining elevation despite frequent flooding,

and the speed-up category indicates that the wetland is gaining elevation under subaerial conditions. Because tides play an important role in examining wetland flooding, we compare the monthly low tide level, mean tide level, and high tide level with the elevation of the wetland surface in the month when SEC data are collected. The wetland response under the monthly low tide level represents the most conservative assessment illustrated in Fig. 4A (the mean and high tide level outcomes are shown in Supplementary Fig. 5).

We find that 87% of the monitoring sites fall in the give-up categories, 5% in the keep-up categories, and the remaining 8% in the catch-up or speed-up categories (Fig. 4B). Thus, only 13% represents what we refer to here as "safe" sites (Fig. 3). Give-up sites are widely distributed across coastal Louisiana with the most vulnerable ones (drowning complete or in progress) clustering in the Chenier Plain and inland portions of the Mississippi Delta (Fig. 4A and Supplementary Fig. 5). In contrast, most of the "safe" sites can be found relatively close to the shoreline in the Mississippi Delta.

## Factors influencing coastal wetland resilience

The impact of sea-level rise on coastal wetlands is complex and varies depending on local conditions and human activities[35]. Model studies have suggested that coastal marshes can keep up with rates of RSL rise >12 mm yr$^{-1}$ [23] or even several times higher under favorable

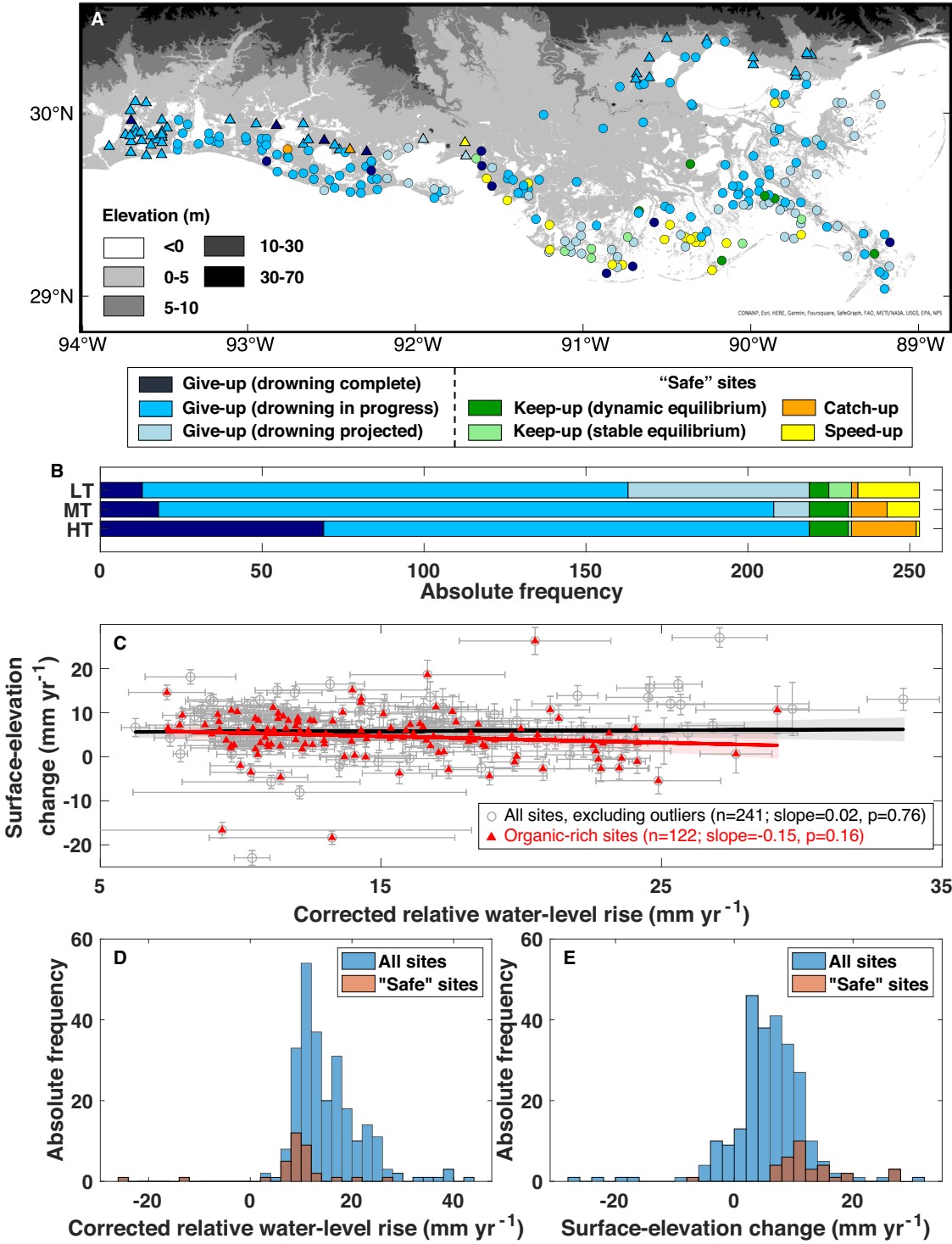

**Fig. 4 | Wetland response to corrected relative water-level (cRWL) change at each monitoring site. A** Wetland response for low tide conditions at 253 monitoring sites (mean tide and high tide conditions are shown in Supplementary Fig. 5). Triangles indicate sites where the Pleistocene basement is <2 m deep ($n = 51$). **B** Comparison of wetland response under low tide (LT), mean tide (MT), and high tide (HT) conditions. **C** Correlation between the rate of cRWL rise and surface-elevation change (SEC). Note that data points with rates of cRWL rise lower than the 2.5th percentile or greater than the 97.5th percentile have been excluded. The smaller subset consists of sites with an organic-matter content >30% (see "Methods"). Error bars represent the standard deviation. **D** Rate of cRWL change for all 253 monitoring sites, with 34 "safe" sites highlighted separately. **E** As in (**D**), but for the rate of SEC.

environmental conditions[8]. This has led to the perception that the vulnerability of marshes may have been overestimated[24]. In contrast, global analyses based on the paleorecord and contemporary in situ surveys have indicated that coastal wetlands are very unlikely to survive when the RSL rise rate exceeds 7 mm yr$^{-1}$ [12], i.e., considerably

lower than the GSL rise rate along the Louisiana coast over the past decade.

Our analysis focuses entirely on the ability of coastal wetlands to adjust vertically to RSL rise. It is well established[36] that these ecosystems can also retreat landward, and some studies have argued that this

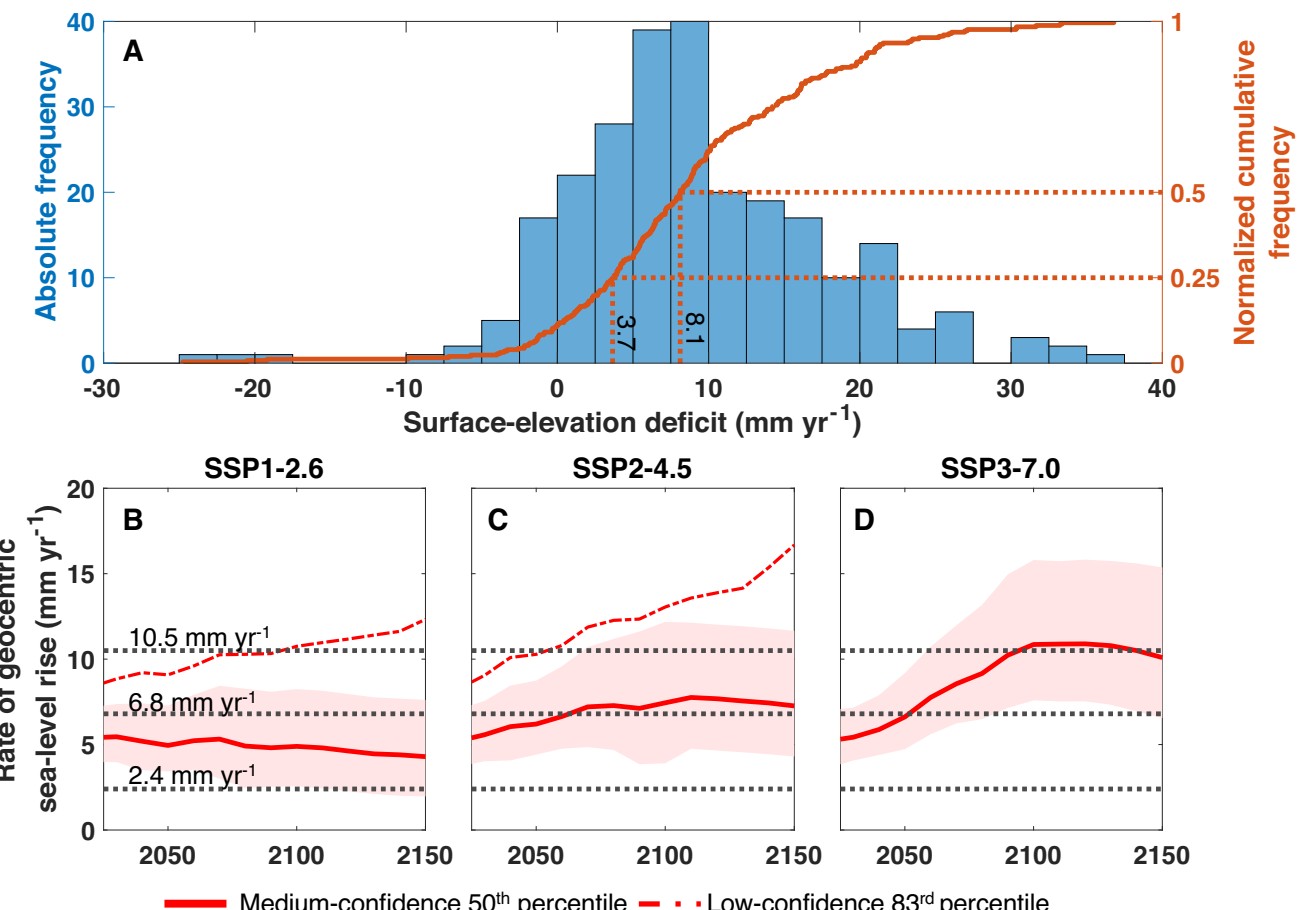

**Fig. 5 | Surface-elevation deficit between 2009 and 2021 and projected rates (low and medium confidence) of geocentric sea-level (GSL) rise**[54–56] **along the Louisiana coast from 2025 to 2150. A** Distribution of surface-elevation deficits for all sites ($n = 253$) and the cumulative distribution curve (negative values indicate a surface-elevation surplus), with values highlighted for the 25th and 50th percentiles. Projected GSL rise rates (with 1σ confidence interval shaded) for SSP1-2.6 (**B**), SSP2-4.5 (**C**), and SSP3-7.0 (**D**). The three dashed lines (black) in (**B–D**) indicate the observed rate of GSL rise (2009–2021), along with rates after subtracting values corresponding to the 25th and 50th percentiles of surface-elevation deficits, respectively.

could result in net areal growth even under pessimistic climate scenarios[37]. However, this neglects the fact that under such scenarios wetlands still must cope with the vertical dimension of rapid RSL rise. In addition, although salt marshes can sometimes migrate landward at the expense of freshwater wetlands, freshwater wetlands often cannot migrate into uplands due to the presence of topographic barriers[38]. Hence, landward migration of coastal wetlands cannot compensate for seaward losses. Coupled modeling of marsh-edge erosion and upland marsh retreat[39] demonstrates how their interplay dictates wetland extent as a function of RSL rise, sediment availability, and the upland slope. While this model study suggested that wetlands can expand under a variety of boundary conditions, a major tipping point was identified when rates of RSL rise exceed a threshold (8–9 mm yr[-1] in[39], although these numbers do not necessarily apply directly to coastal Louisiana). Under such conditions, widespread inundation is predicted with a rapid reduction in wetland extent, similar to what has been observed for the early Holocene in the Mississippi Delta[40]. We note that wetland sites in our study area which would constitute the nucleus for landward retreat (i.e., those farthest landward that abut gently sloping uplands; triangles in Fig. 4A and Supplementary Fig. 5) dominantly (96%) fall in the give-up category. This is consistent with the model results[39] predicting that marsh interiors are particularly vulnerable, with higher resilience near the open coast (as seen in Fig. 4A), corresponding to the inundation scenario where drowning commences in the marsh interior until the seaward edge ultimately jumps landward. This results in a fringing marsh with a much reduced footprint[41].

Climate change and elevated $CO_2$ levels may enhance vertical accretion in coastal wetlands by biological feedbacks, another potential mechanism through which wetlands might counterbalance rising sea levels in the future[23,24,42]. An experimental study conducted in Chesapeake Bay, USA, demonstrated how increased $CO_2$ can stimulate vertical accretion but showed a decline in productivity once the rate of RSL rise surpasses 7 mm yr[-1 43]. We compare the rates of cRWL change and SEC at all monitoring sites (excluding a small number of outliers; $n = 12$) in addition to a subset of sites ($n = 122$) with high organic-matter content, where biological feedbacks are more likely to dominate vertical accretion. Our data show no correlation for either case (Fig. 4C). In other words, we see no evidence for the ecogeomorphic feedbacks proposed by previous studies[23,24,42], consistent with observations over the past three decades from the Everglades, Florida[44]. Despite the high median rate of cRWL rise in coastal Louisiana, 13% of our sites are not yet in give-up mode. These "safe" sites experience a median cRWL rise of 9.7 mm yr[-1], i.e., much lower than the overall median (Fig. 4D). This subset of resilient wetland sites also features SEC rates that are roughly twice as high as the median for the entire data set (11.1 vs. 6.0 mm yr[-1]; Fig. 4E). Given our finding that sites with lower rates of cRWL rise tend to see higher SEC rates, we postulate that the rates of cRWL rise in our study area are well above those where biological productivity benefits from increased flooding. Therefore, we suggest that future studies to understand ecogeomorphic feedbacks be conducted in areas with lower rates of RSL rise, a larger tidal amplitude, and/or semidiurnal or mixed tidal regimes.

## Vulnerability of coastal wetlands under future sea-level projections

The median surface-elevation deficit for all monitoring sites during the study period is 8.1 mm yr⁻¹ (Fig. 5A), suggesting that even with a considerably lower rate than the observed GSL rise, widespread wetland collapse is likely to occur. Subtracting the median surface-elevation deficit from the observed rate of GSL rise yields 2.4 mm yr⁻¹, a condition where about half of the monitoring sites would be able to track GSL rise. This is consistent with earlier inferences[20] that adopted a GSL rise rate of only 2 mm yr⁻¹ and found about half of the monitoring sites to be in deficit. We also examine the outcome for the 25th percentile (Fig. 5A) which corresponds to a surface-elevation deficit of 3.7 mm yr⁻¹. A similar subtraction yields a value of 6.8 mm yr⁻¹, suggesting that under such a rate of GSL rise about 75% of sites would be in deficit.

According to the Sixth Assessment Report of the Intergovernmental Panel on Climate Change (IPCC-AR6)[45], with policies currently in place, we are approximately following Shared Socioeconomic Pathway (SSP) 2–4.5. Projections of GSL rise along the Louisiana coast indicate that even under SSP1-2.6 (which would require Paris Agreement objectives to be achieved), the rate of GSL rise is very likely to exceed 2.5 mm yr⁻¹, putting at least half of the sites in danger of drowning (Fig. 5B). Under SSP2-4.5, GSL rates are projected to surpass 7 mm yr⁻¹ by 2070 (Fig. 5C). As a result, it is plausible that ~75% of the wetlands will lack the resilience necessary to withstand rising sea level by 2070 under the present climate scenario. Under SSP3-7.0, it is more likely than not that the rate of GSL rise observed over the past decade will be reached by the end of this century, with ~90% of wetlands drowning as a result (Fig. 5D). It is worth noting that these GSL projections are based on medium-confidence processes; without consideration of low-confidence processes (high magnitude/low probability non-linear ice-sheet responses[7]). Recent studies have reported increasing contributions to GSL rise from both Greenland and Antarctica[46–48], indicating that high-impact, low-confidence processes cannot be ruled out. The corresponding low-confidence GSL rise projections for SSP1-2.6 and SSP2-4.5 (83rd percentile) exhibit higher rates and magnitudes than the median projections for SSP3-7.0. As a result, ~90% drowning of wetlands may occur as early as 2090 and 2060, respectively under these scenarios (Fig. 5B, C).

As mentioned before, our focus herein is on the vertical adjustment of coastal wetlands with an emphasis on low-tide conditions (Fig. 4A). It is also important to note that the RSETs were deliberately established in relatively intact wetlands to enable long-term monitoring, i.e., land loss due to marsh-edge erosion is generally not captured by our analysis. Furthermore, it is reasonable to expect that multi-decadal cycles due to ocean dynamic changes will continue to be superimposed on the climate-driven global sea-level rise[21], i.e., periods with rates of GSL rise higher than in the past decade may well occur before 2070. Combined, all these factors suggest that our analysis may underestimate wetland vulnerability. The aforementioned threshold rate of 7 mm yr⁻¹ [12] corresponds to a condition where about 75% of our monitoring sites are in deficit. Although we are reluctant to convert this number into precise rates of wetland loss, within a time window of <50 years this would translate into loss rates well above anything that has been observed in the past century.

The unusual exposure of the Louisiana coast to accelerated sea-level rise over the past decade provides a unique opportunity to time travel to conditions not expected until later in this century. As such, this study offers empirical evidence of the transformation of a heavily human-influenced landscape due to climate change by means of a globally unprecedented monitoring system. This climate-impact experiment shows that widespread collapse of coastal wetlands in this area may be expected by 2070 or earlier. While this outcome may not be entirely avoidable, climate mitigation along with major restoration efforts by means of sediment diversions[49] could delay wetland drowning and allow for more time to prepare for this large-scale coastal transformation.

## Methods

### Geocentric sea-level (GSL) data

We obtained GSL data from coastal satellite altimetry, a product that is freely available from the SEANOE repository (https://doi.org/10.17882/74354). This includes monthly GSL time series (with annual and semi-annual cycles removed) from January 2002 to December 2019 at 756 satellite tracks (satellite altimetry stations) from 20 km offshore to the shoreline. This dataset was originally provided by the Jason 1, 2, and 3 missions and reprocessed by computing high-resolution along-track altimetry ranges, applying an adaptive leading-edge subwaveform retracking method, plus geophysical and environmental corrections and re-estimating the inter-bias missions at the regional scale[26]. Four satellite altimetry virtual stations (Chenier, Marsh Island, Breton Sound, and Mississippi Sound) were used in this study, providing rates of GSL change at each data point within the tracks (Supplementary Fig. 1). GSL change at the Grand Isle tide gauge was derived by using the nonlinear vertical land motion (VLM) correction from[21] and subsequently compared with the data points at the four satellite altimetry virtual stations (Supplementary Fig. 2).

### Relative water-level (RWL) data

RWL data (2009–2021) are available from water-level gauges at 382 monitoring sites throughout coastal Louisiana by means of the Coastwide Reference Monitoring System (CRMS). Hourly measurements were converted to daily mean values. The data are provided with respect to the NAVD 88 (Geoid 12 A) reference system (geodetic surveys of the water-level gauges were performed in 2014) and collected from open water bodies like bays, bayous, or ponds that are hydrologically connected to the adjacent wetland where surface-elevation data are collected (see below and Supplementary Fig. 3). The water-level gauges are attached to wooden posts, typically with a 4–5 m installation depth below the nearby wetland surface (Supplementary Fig. 4). This setup indicates that besides the geocentric water level, VLM (i.e., subsidence) below the post's installation depth is also captured by these measurements. Hence, they provide RWL measurements. Monthly mean values of RWL were calculated based on daily means, considering only the months with a daily data completeness exceeding 70%. Following this, sites with monthly mean data completeness below 70% were excluded, resulting in a final selection of 325 sites. To remove seasonality from the monthly RWL data, we calculated the long-term average over each calendar month (after detrending the raw RWL data), and then subtracted the long-term average from the corresponding month over the entire study period.

### Surface-elevation change (SEC) data

SEC is monitored by the aforementioned CRMS network by means of the rod surface-elevation table (RSET). SEC is measured biannually with vertical pins that slide through a horizontal steel arm that is attached to a vertical rod driven to ~20 m depth (Supplementary Fig. 4). Geodetic surveys of the RSETs (top of the rod) were performed in 2014, enabling the wetland surface elevation to be converted to the NAVD 88 (Geoid 12 A) reference system with the known distance between the top of the rod and the horizontal steel arm. Again, we excluded sites with <70% of data completeness, resulting in 253 sites with both SEC and RWL data.

### Statistical analysis

Linear trends of monthly GSL, RWL, and biannual SEC data were estimated using ordinary least squares regression. Pairwise correlation analyses were conducted between the residuals (detrended data) of GSL change at Grand Isle and the satellite altimetry-derived GSL changes from 2009 to 2019, as well as between the residuals of GSL

change at Grand Isle and the RWL change at the water-level gauges from 2009 to 2021.

To test the statistical significance of the linear trends in monthly GSL and RWL data, and to obtain correlation coefficients, we applied Monte Carlo simulations by assuming the residuals of GSL and RWL data can be explained by an autoregressive process of the order 1 (AR1), and 10,000 red noise time series were generated to simulate the residuals[50]. We used the 10,000 synthetic red noise series to calculate the significance of observed linear trends and correlation coefficients. Due to the limited length of SEC records for estimating the AR1 parameters, we generated 10,000 white noise time series instead by assuming the residuals are temporally uncorrelated for all SEC records.

## Vertical land motion (VLM) correction

All RWL and SEC measurements are affected by VLM. However, the installation depths of the RSETs and water-level gauges are substantially different, resulting in different behaviors between the two instruments. If both instruments are installed in the largely compaction-free Pleistocene basement, no differential VLM correction is required as no shallow sediment compaction is captured by the data, and deep subsidence affects both instruments equally (Supplementary Fig. 4). However, when one or both instruments do not penetrate into the Pleistocene basement (i.e., they are "floating" in the compaction-prone Holocene strata; Supplementary Fig. 4), correction for differential VLM is necessary. Therefore, we extracted installation depths of the RSETs (https://www.lacoast.gov/crms_viewer/Map/CRMSViewer) and used the depth of the Holocene-Pleistocene (HP) interface[51] to determine whether VLM correction is required. For RSETs with unknown installation depth (n = 17), we used a value of 20 m which approximates the mean installation depth for all RSETs in this study. For RSETs at locations with unknown depths of the HP interface (n = 14; all in the western Chenier Plain) we used the nearest neighbor values.

The mean shallow subsidence rate for the top ~20 m in coastal Louisiana has been reported as $6.8 \pm 7.9$ mm yr$^{-1}$ [20]. Since most compaction occurs in the top 1–3 m[52] and the installation depth of the water-level gauges is approximately 4–5 m, the differential VLM correction between the water-level gauges and the RSETs is likely smaller than the shallow subsidence rate. Here, we assume that the VLM correction between RSETs and nearby water-level gauges follows a normal distribution, where 1 and 4 mm yr$^{-1}$ correspond to the 5th and 95th percentiles, respectively. We also assumed that these corrections remain constant throughout the study period. We then randomly selected 10,000 VLM corrections from this distribution and subtracted them from the 10,000 linear RWL trends generated previously for each site.

In addition to the linear trends, the RWL elevation, measured with respect to the installation depth of the water-level gauges, is also influenced by VLM. To facilitate further analyses, we applied a VLM correction to the monthly RWL elevations (raw data without seasonal correction) from 2009 to 2021. Since the water-level gauges were surveyed in 2014 (requiring no VLM correction for that year), we adjusted the RWL elevations after 2014 by subtracting the cumulative elevation changes (rate of VLM multiplied by the number of years from 2014). Conversely, we adjusted the RWL elevations before 2014 by adding the cumulative elevation changes. Ultimately, both wetland and water-surface elevations are referenced to the installation depths of the RSETs under the NAVD 88 datum (surveyed in 2014).

## Wetland response classification

The wetland response to RWL change is classified by examining the relationship between the cRWL (RWL after VLM correction) and SEC data, as well as the amount of flooding (Fig. 3). At each site, we determined the monthly cRWL as the mean tide, while calculating low tide and high tide by subtracting or adding half the annual tidal

amplitude from the mean tide, respectively. Then, for each site, we compared the rates of cRWL change and SEC using the 10,000 Monte Carlo simulations. Subsequently, we assessed the flooding condition by comparing the wetland surface elevation with the elevation of low tide (Fig. 4A), mean tide (Supplementary Fig. 5A), and high tide (Supplementary Fig. 5B) during the same month when the SEC data was collected. The classification details are as follows:

- Give-up (drowning complete): the cRWL change rate is likely (≥66%) to be higher than the SEC rate, and the wetland is very likely (≥90%) to be flooded during the study period.
- Give-up (drowning in progress): the cRWL change rate is likely (≥66%) to be higher than the SEC rate, and the probability of wetland flooding during the study period ranges from 10 to 90%.
- Give-up (drowning projected): the cRWL change rate is likely (≥66%) to be higher than the SEC rate, but the wetland is very unlikely (≤10%) to be flooded during the study period.
- Keep-up (dynamic equilibrium): the cRWL change rate is about as likely as not (33–66%) to be higher than the SEC rate, and the probability of wetland flooding during the study period is >10 %.
- Keep-up (stable equilibrium): the cRWL change rate is about as likely as not (33–66%) to be higher than the SEC rate, and the wetland is very unlikely (≤10%) to be flooded during the study period.
- Catch-up: the cRWL change rate is unlikely (≤33%) to be higher than the SEC rate, and the wetland is more likely than not (>50%) to be flooded during the study period.
- Speed-up: the cRWL change rate is unlikely (≤33%) to be higher than the SEC rate, and the probability of wetland flooding is ≤50% during the study period.

In addition to categorizing the wetland response, we calculated the surface elevation deficit (the rate difference between cRWL change and SEC) of the wetlands over 10,000 simulations for each site.

## GSL projections

Future rates of GSL change are available from the IPCC-AR6 Sea Level Projections (http://zenodo.org/record/6382554). Along the Louisiana coast (Lat: 28.8° to 30.6°; Lon: −94° to −88.8°), there are 16 locations with projected rates of GSL rise until 2150 with 10-year increments. We used the mean value by averaging the data from all 16 locations at each 10-year increment until 2150. The 1σ confidence intervals of the projected rates are also provided in the dataset.

## Organic-matter content

Soil properties, including organic-matter content (%) for the top 24 cm, are available from CRMS (https://cims.coastal.louisiana.gov/monitoring-data/) based on shallow cores that were collected at 239 sites in 2018. The mean organic-matter content was calculated by averaging the values for the six 4-cm-increments analyzed in each core. Of the 253 monitoring sites used in the analysis of the SEC-RWL change relationship, only 223 have soil core data, and 122 of these sites are organic-rich (organic-matter content >30%).

## Data availability

The raw RWL and SEC data that support the findings of this study can be retrieved from the Coastal Information Management System database (https://cims.coastal.louisiana.gov/monitoring-data/). Source data are provided with this paper.

## Code availability

The data and code for the analysis of wetland response classification have been deposited in the ZENODO database (https://doi.org/10.5281/zenodo.10543670)[53]. Codes to produce the figures are available from the corresponding author upon request.

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

## Acknowledgements

This study was supported by the U.S. Department of the Treasury through the Louisiana Coastal Protection and Restoration Authority's Center of Excellence Research Grants Program under the Resources and Ecosystems Sustainability, Tourist Opportunities, and Revived Economies of the Gulf Coast States Act of 2012 (RESTORE Act) (Award No. 1 RCEGR260007-01-00). The statements, findings, conclusions, and recommendations are those of the authors and do not necessarily reflect the views of the Department of the Treasury. S.D. acknowledges NASA grant no. 80NSSC20K1241 and David and Jane Flowerree for their endowment funds. Fieldwork was conducted and data was provided through the Coastwide Reference Monitoring System (CRMS), for which we are grateful. We thank the projection authors for developing and making the sea-level rise projections available, multiple funding agencies for supporting the development of the projections, and the NASA Sea Level Change Team for developing and hosting the IPCC AR6 Sea Level Projection Tool. We also thank José Silvestre (Tulane University); Leigh Anne Sharpe (Coastal Protection and Restoration Authority of Louisiana); Anjali Fernandes (Denison University); Paul Heinrich (Louisiana Geological Survey); Daniel Friess (Tulane University); and members of the Quaternary Research Group at Tulane University for their comments, discussion, and support.

## Author contributions

Conceptualization: G.L., T.E.T. Methodology: G.L., T.E.T., S.D. Visualization: G.L. Funding acquisition: T.E.T. Supervision: T.E.T., S.D. Writing—original draft: G.L. Writing—review & editing: G.L., T.E.T., S.D.

## Competing interests

The authors declare no competing interests.
