## [Peer Review File · Nature Communications]

Real-world time-travel experiment shows ecosystem collapse due to anthropogenic climate changeREVIEWER COMMENTS

Reviewer #1 (Remarks to the Author):

This investigation capitalizes on the accelerated sea-level rise (> 10 mm/yr) in the Gulf of Mexico since ~ 2010 (17,18). I consider it a novel idea to capitalize on the current rate of sea-level rise to undertake this experiment. While this acceleration is believed to be caused by oceanographic processes, it provides an opportunity to investigate wetlands response to rates of rise that are projected for this century. Specifically, can vertical accretion of coastal wetlands keep pace with rates of rise projected for the next few decades.

The manuscript is well written and well organized. The figures are legible and the captions clear and succinct.

The authors conclude that, given the median surface-elevation deficit for all monitoring sites of 8.0 mm/yr, widespread wetlands loss is "likely to occur". They further quantify this loss as $\sim 75\%$ by 2070 under the present IPCC climate scenario and $\sim 90\%$ by the end of the century, perhaps faster given the assumption that decadal oscillations in sea level driven by oceanographic processes will continue into the future. These results are consistent with those of previous studies, including works by the authors, and with the latest NOAA projections, as conveyed by the online interactive NOAA dashboard. For example, the NOAA dashboard shows a similar magnitude on coastal inundation to that predicted in this manuscript, with $\sim 75\%$ loss by the year 2050 with an approximately 30 cm sea-level rise. The two main uncertainties for both models are (1) actual greenhouse gas concentrations over the next few decades and (2) ice sheet contributions to sea-level rise. The recent literature on both Greenland and Antarctica indicate that their contributions are increasing and are on the verge of dominating sea-level rise. It is important to point out that the inundation scenarios for south Louisiana will remain uncertain due to these uncertainties.

It would be useful if Figure 1 included some relief to show that topographic constraints to upland migration are far from the coast.

In summary, the novel approach of this research will interest readers and the results do indeed provide a glimpse of what the future holds. However, the authors should be more forthcoming in stating that the main results from this work are consistent with previous studies. In my opinion, this does not detract from the importance of their work, which is that big changes are in store for the Louisiana coast in just a few decades.

Reviewer #2 (Remarks to the Author):

The manuscript entitled "First real-world time-travel experiment shows ecosystem collapse due to climate change" by Li and colleagues provides a solid dataset examining the drivers and rates of relative sea-level change across the Mississippi Delta region. It provides a nice incremental advance of their earlier work by Jankowski et al. (2017). Based on their analysis of current changes, they provide an alarming prediction for the collapse of the region's expansive wetlands, similar to other previous studies (e.g. Day et al., 2000; Morton et al., 2005; Blum and Tornqvist, 2010; Tornqvist et al., 2020), by the year 2070. It is a very careful and thorough study, although I do have a concern about one of their assumptions, but I wonder if the conclusions are "novel" enough to warrant publication in such a high impact journal – many studies have shown the Delta is in trouble.

Title: I wonder if "ecosystem" is the best term. Yes, the definition includes their physical environment but an ecosystem is foremost a biological community of interacting organisms. This paper really focuses on the physical environment and not the interacting organisms. Maybe "wetland" instead of "ecosystem?"

Although the methods are for the most part, solid, I wonder how ignoring "deep subsidence" – line 552-553 might impact this analysis. From my understanding of your methods, "deep subsidence" isn't even that "deep" as don't your rods only go down to 20 m or so? Your cartoon shows "non-compressible" Pleistocene sediments, which although might be non-compressible, they still are loading the lithosphere. The MIS5 surfaces are 10's of m below the surface in many of the locations suggesting reasonably high rates of "deep subsidence" in at least some parts of the delta – particularly your more seaward sites. Yu et al. (2012) show only ~0.15 mm/yr of subsidence but those estimates are all landward of New Orleans. Subsidence rates (and many of your locations) are seaward of that line and likely experience much higher rates of "deep subsidence" driven by loading of the lithosphere. At the least, deeper sediment loading should be discussed.

Other minor comments:

Line 67: "evidence" – do you mean "estimates" or "work"?

Line 83: remove "s" from "feedback"

Line 120: "positive trend" – do you mean RWL rise?

Line 124 – need a reference...

Line 131-132: the reference to "prolonged floods" seems weird to introduce in a sentence that starts with "In summary..." when prolonged floods is not mentioned anywhere else in the paragraph

Line 136: "RSL, i.e. GSL plus other factors including subsidence" – I think RSL includes other things besides just GSL and subsidence – e.g. tectonics...

Line 174: This reference is to Mangroves, not marshes, can you find one for marshes?

Line 203: How many sites did you exclude – please give the number

Line 205: Please provide the statistics showing no correlation...

Line 255: But isn't the rate already exceeding 2.5 mm/yr?

Line 543: Remove extra "aaa"

Reviewer #3 (Remarks to the Author):

This manuscript by Li et al. examines coastal Louisiana wetland responses to especially rapid water level changes since 2010. The results are quite noteworthy and novel, namely that a detailed analysis of 253 sites across coastal Louisiana show that the vast majority are unable to keep pace with rapid rates of water level rise. As the authors point out, understanding wetland response to rapidly rising sea-level is of the utmost importance, especially in coastal Louisiana. To my knowledge, this is the first study of its kind, and was very cleverly put together. Furthermore, this is a very well written manuscript that contains robust methodologies, clean figures, and detailed statistical analyses to support the dataset and conclusions drawn. There is sufficient detail for the work to be reproduced. This work will be a significant contribution to the field of wetland/marsh studies both in terms of field- and modeling -based approaches, and does a good job framing the importance, context, and conclusions.

Overall, I have few comments as this is a very polished study in its current form. One important observation that struck me, I would like to see the authors comment on the minimum time period that the classification schemes are relevant. For the vast majority of the dataset, the schemes presented are obvious and intuitive. To be clear, I do believe the authors framing the argument as using these short-term observations as a window into the future is entirely appropriate and meaningful. However, I was left wondering if one examines shorter time periods within the dataset, how would the classification be different? Using the examples in figure 3, in the "catch-up" regime, it appears that from 2009-2016, this might have been classified as one of the "give-up" scenarios. Yet from 2016-2019, it appears the wetland surface accelerated its elevation growth to fall overall within the "catch-up" regime. In other words, how long do they need to be in a "give-up" scenario before it is clear it won't shift back to a "catch-up"? I ask this question because, as the authors point out, these high RWL

increases observed during this period are likely temporary at least in the very short-term. If that happens, what will be the corresponding wetland response over short-time scales? I certainly do not deny that drowning is inevitable over even a decade of a continuous "give-up" scenario, but since this study is focused on short timescales, I believe this to be a relevant question and perhaps the only currently available dataset to answer this question. This is obviously relevant for management-related strategies.

Along these lines, it would also be interesting to see shorter time scale wetland surface elevation gain rates and their variability.

Minor comments:

Figure 3: It would be helpful to see these CRMS station examples identified somewhere on a map. This would also help visualize the observed variations in tidal range and wetland surface elevation gains. I would also recommend adding more spaces in the text on the left cartoons, as it seems a bit close together.

Figure 4A: I recommend changing the color scheme for the give-up scenarios, perhaps by adding a symbol in the dot. They are a bit hard to differentiate as is.

Figure 4C: Can error bars be included with these data points?

Figure 5B: I recommend putting each IPCC scenario label also in each panel box for B-D

Reviewers' original comments are included below in black text

Manuscript Authors' responses to Reviewers' comments are included below the relevant comment in blue, bold text.

Additional edits to the manuscript that were not prompted by a Reviewer comment are described in a bulleted list in green, bold text at the end of this document.

Reviewers' comments:

Reviewer #1 (Remarks to the Author):

This investigation capitalizes on the accelerated sea-level rise (> 10 mm/yr) in the Gulf of Mexico since ~ 2010 (17,18). I consider it a novel idea to capitalize on the current rate of sea-level rise to undertake this experiment. While this acceleration is believed to be caused by oceanographic processes, it provides an opportunity to investigate wetlands response to rates of rise that are projected for this century. Specifically, can vertical accretion of coastal wetlands keep pace with rates of rise projected for the next few decades.

The manuscript is well written and well organized. The figures are legible and the captions clear and succinct.

We thank the reviewer for the encouraging words.

The authors conclude that, given the median surface-elevation deficit for all monitoring sites of 8.0 mm/yr, widespread wetlands loss is "likely to occur". They further quantify this loss as ~75% by 2070 under the present IPCC climate scenario and ~ 90% by the end of the century, perhaps faster given the assumption that decadal oscillations in sea level driven by oceanographic processes will continue into the future. These results are consistent with those of previous studies, including works by the authors, and with the latest NOAA projections, as conveyed by the online interactive NOAA dashboard. For example, the NOAA dashboard shows a similar magnitude on coastal inundation to that predicted in this manuscript, with ~75% loss by the year 2050 with an approximately 30 cm sea-level rise.

We thank the reviewer for this comment and add the NOAA sea-level report to our references. While the NOAA inundation dashboard shows a similar magnitude of coastal inundation, it is essential to clarify that the underlying principles of our model differ significantly. The inundation map primarily addresses comparisons between the mean higher high water level and the stable land surface elevation, whereas our study focuses on dynamic wetland surface elevation changes. The inundation map from the NOAA dashboard can be attributed primarily to the low-lying nature of Louisiana's coastal areas and the impact of sea-level rise. However, our study suggests that, besides these two factors, the inability of the wetlands to gain elevation with sea-level rise is also important.

The two main uncertainties for both models are (1) actual greenhouse gas concentrations over the next few decades and (2) ice sheet contributions to sea-level rise. The recent literature on

both Greenland and Antarctica indicate that their contributions are increasing and are on the verge of dominating sea-level rise. It is important to point out that the inundation scenarios for south Louisiana will remain uncertain due to these uncertainties.

We thank the reviewer for this comment and agree that these future uncertainties should be discussed. Therefore, we briefly add information about the increasing contribution to the GSL rise from the Greenland and Antarctica ice sheets. We also include 83rd low-confidence GSL projections from IPCC AR6 along the Gulf Coast under SSP1-2.6 and SSP2-4.5 in updated Figs. 5B and C and point out that the widespread drowning of wetlands (~90%) could occur earlier (2090 and 2060, respectively) by considering these low-confidence processes. Below are the added sentences in the main text, in italics:

“It is worth noting that these GSL projections are based on medium-confidence processes; without consideration of low-confidence processes (high magnitude/low probability non-linear ice-sheet responses⁶). Recent studies have reported increasing contributions to GSL rise from both Greenland and Antarctica ⁴⁴⁻⁴⁶, indicating that high-impact, low-confidence processes cannot be ruled out. The corresponding low-confidence GSL rise projections for SSP1.2.6 and SSP2-4.5 (83rd percentile) exhibit higher rates and magnitudes than the median projections for SSP3-7.0. As a result, ~90% drowning of wetlands may occur as early as 2090 and 2060, respectively under these scenarios (Figs. 5B and C).”

It would be useful if Figure 1 included some relief to show that topographic constraints to upland migration are far from the coast.

We thank the reviewer for this comment. Because the legend in Figure 1 can block some topographic features, we then add the topographic constraints in Figure 4A and Extended Data Figure 5.

In summary, the novel approach of this research will interest readers and the results do indeed provide a glimpse of what the future holds. However, the authors should be more forthcoming in stating that the main results from this work are consistent with previous studies. In my opinion, this does not detract from the importance of their work, which is that big changes are in store for the Louisiana coast in just a few decades.

We thank the reviewer for this encouraging evaluation and agree that our findings are broadly consistent with previous work, not least our own previous work. Nevertheless, we still consider this study a major advance, because unlike Jankowski et al. (2017) it includes very detailed data on GSL rise (based on recently published coastal altimetry data) and associated RWL rise throughout the study area (the previous study simply assumed a highly conservative rate of 2 mm/yr of water-level rise for the entire region). Furthermore, we now include projections when drowning conditions could occur in the future based on this novel dataset.

We have added a sentence in the final paragraph to stress the fact that our study is novel, by reiterating that the kind of “time-travel experiment” that we have been able to carry out is unique, given the scope and extent of the monitoring program used. Below are the added sentences in the main text, in italics:

“As such, this is the first study to offer empirical evidence of the transformation of a heavily human-influenced landscape due to climate change by means of a globally unprecedented monitoring system.”

Reviewer #2 (Remarks to the Author):

The manuscript entitled “First real-world time-travel experiment shows ecosystem collapse due to climate change” by Li and colleagues provides a solid dataset examining the drivers and rates of relative sea-level change across the Mississippi Delta region. It provides a nice incremental advance of their earlier work by Jankowski et al. (2017). Based on their analysis of current changes, they provide an alarming prediction for the collapse of the region’s expansive wetlands, similar to other previous studies (e.g. Day et al., 2000; Morton et al., 2005; Blum and Tornqvist, 2010; Tornqvist et al., 2020), by the year 2070. It is a very careful and thorough study, although I do have a concern about one of their assumptions, but I wonder if the conclusions are “novel” enough to warrant publication in such a high impact journal – many studies have shown the Delta is in trouble.

We thank the reviewer for their critical evaluation. As we outline in detail in our answer to Reviewer 1, we would argue that our study is novel in several respects. For example, we believe that a dataset anchored by hundreds of time series of RWL change and associated wetland response is globally unprecedented. More broadly, we are not aware of any empirical climate impact studies that offer such a comprehensive picture of conditions that are not expected until later in this century. We have added a sentence in the final paragraph to stress the fact that our study is novel, by reiterating that the kind of “time-travel experiment” that we have been able to carry out is unique, given the scope and extent of the monitoring program used. Below are the added sentences in the main text, in italics:

“As such, this is the first study to offer empirical evidence of the transformation of a heavily human-influenced landscape due to climate change by means of a globally unprecedented monitoring system.”

In addition, we also recognize the early land loss studies in Louisiana by adding two references in line 61. The references are shown below

Craig, N., Turner, R. E. & Day, J. W. Land loss in coastal Louisiana (USA). *Environmental Management* 3, 133-144 (1979).

Gagliano, S. M., Meyer-Arendt, K. J. & Wicker, K. M. Land loss in the Mississippi River deltaic plain. *Gulf Coast Association of Geological Societies Transactions* 31, 295-300 (1981).

Title: I wonder if “ecosystem” is the best term. Yes, the definition includes their physical environment but an ecosystem is foremost a biological community of interacting organisms. This paper really focuses on the physical environment and not the interacting organisms. Maybe “wetland” instead of “ecosystem?”

While we understand where the reviewer is coming from, and we also recognize that our study encompasses more than just one ecosystem (different marsh types as well as swamps), we have chosen not to change this term as suggested. The point we are trying to make is that climate impact studies based on direct field observations at the spatiotemporal scale of our case have not yet been carried out for any type of ecosystem. Hence, “wetland” would be too narrow a term. Also, even though we do not explicitly discuss communities of interacting organisms, it is implicit from our findings that the drowning of marshes will cause the demise of these communities.

Although the methods are for the most part, solid, I wonder how ignoring “deep subsidence” – line 552-553 might impact this analysis. From my understanding of your methods, “deep subsidence” isn’t even that “deep” as don’t your rods only go down to 20 m or so? Your cartoon shows “non-compressible” Pleistocene sediments, which although might be non-compressible, they still are loading the lithosphere. The MIS5 surfaces are 10’s of m below the surface in many of the locations suggesting reasonably high rates of “deep subsidence” in at least some parts of the delta – particularly your more seaward sites. Yu et al. (2012) show only ~0.15 mm/yr of subsidence but those estimates are all landward of New Orleans. Subsidence rates (and many of your locations) are seaward of that line and likely experience much higher rates of “deep subsidence” driven by loading of the lithosphere. At the least, deeper sediment loading should be discussed.

We recognize the significance of this comment and agree that the “deep subsidence” rates increase seaward. However, please note that in our analysis we do consider the deep subsidence, which is already included in the RWL data.

To avoid confusion, we removed the sentence ‘with deep subsidence ignored’ and replaced it with ‘surveyed in 2014’ in line 585. This is only for the elevation comparison. For the direct elevation comparison between the water surface and wetland surface, the vertical datum is NAVD 88, Geoid 12A, which was surveyed back in 2014. Since we already applied the VLM correction to the water-level gauge data, both SEC and cRWL (VLM-corrected RWL) are relative to the same reference point at each site which is ~20 m (the base of the RSET rods) and can be compared directly.

Other minor comments:

Line 67: “evidence” – do you mean “estimates” or “work”?

We changed “limited evidence” to “scarcity of studies on regional...”

Line 83: remove “s” from “feedback”

Revised as reviewer suggested.

Line 120: “positive trend” – do you mean RWL rise?

Yes, we replaced the positive trend with RWL rise and rewrote the sentence.

Line 124 – need a reference...

We added following reference here:

Russell, R.J. Physiography of Lower Mississippi River Delta. Louisiana Department of Conservation Geological Bulletin 8, 3-199 (1936).

Line 131-132: the reference to “prolonged floods” seems weird to introduce in a sentence that starts with “In summary...” when prolonged floods is not mentioned anywhere else in the paragraph

We rewrote this part as follows:

“This is the area where monitoring sites exhibit lower correlation coefficients between RWL change and GSL change at the Grand Isle tide gauge (Fig. 2B). In the Mississippi Delta, a higher correlation is observed closest to the coast, suggesting a decreasing ocean influence farther inland. At the birdfoot delta, slightly lowered correlation coefficients may be related to elevated water levels during prolonged floods²⁷ (Fig. 2B). In summary, RWL changes across the Louisiana coast show that 97.2% of sites exhibit a statistically significant correlation ($p < 0.001$) with GSL change at the Grand Isle tide gauge, with a median correlation coefficient of 0.74 (Fig. 2B). Hence, GSL rise is the main factor driving RWL rise in coastal Louisiana, except in the highly impounded Chenier Plain²⁶ and near river mouths.”

Line 136: “RSL, i.e. GSL plus other factors including subsidence” – I think RSL includes other things besides just GSL and subsidence – e.g. tectonics...

We changed this to ‘GSL plus vertical land motion’.

Line 174: This reference is to Mangroves, not marshes, can you find one for marshes?

This reference covers both tidal marshes and mangroves (along with coral reef islands).

Line 203: How many sites did you exclude – please give the number

We have added the number of outliers in the main text, and in the legend of Figure 4C.

Line 205: Please provide the statistics showing no correlation...

We have added this information in the legend of Figure 4C.

Line 255: But isn’t the rate already exceeding 2.5 mm/yr?

Yes, it is. We want to compare our results with the previously used estimate by Jankowski et al. (2017), which was 2 mm/yr of GSL rise.

Line 543: Remove extra “aaa”

We removed the typo.

Reviewer #3 (Remarks to the Author):

This manuscript by Li et al. examines coastal Louisiana wetland responses to especially rapid water level changes since 2010. The results are quite noteworthy and novel, namely that a detailed analysis of 253 sites across coastal Louisiana show that the vast majority are unable to keep pace with rapid rates of water level rise. As the authors point out, understanding wetland response to rapidly rising sea-level is of the utmost importance, especially in coastal Louisiana. To my knowledge, this is the first study of its kind, and was very cleverly put together. Furthermore, this is a very well written manuscript that contains robust methodologies, clean figures, and detailed statistical analyses to support the dataset and conclusions drawn. There is sufficient detail for the work to be reproduced. This work will be a significant contribution to the field of wetland/marsh studies both in terms of field- and modeling -based approaches, and does a good job framing the importance, context, and conclusions.

We thank the reviewer for the encouraging words.

Overall, I have few comments as this is a very polished study in its current form. One important observation that struck me, I would like to see the authors comment on the minimum time period that the classification schemes are relevant. For the vast majority of the dataset, the schemes presented are obvious and intuitive. To be clear, I do believe the authors framing the argument as using these short-term observations as a window into the future is entirely appropriate and meaningful. However, I was left wondering if one examines shorter time periods within the dataset, how would the classification be different? Using the examples in figure 3, in the “catch-up” regime, it appears that from 2009-2016, this might have been classified as one of the “give-up” scenarios. Yet from 2016-2019, it appears the wetland surface accelerated its elevation growth to fall overall within the “catch-up” regime. In other words, how long do they need to be in a “give-up” scenario before it is clear it won’t shift back to a “catch-up”? I ask this question because, as the authors point out, these high RWL increases observed during this period are likely temporary at least in the very short-term. If that happens, what will be the corresponding wetland response over short-time scales? I certainly do not deny that drowning is inevitable over even a decade of a continuous “give-up” scenario, but since this study is focused on short timescales, I believe this to be a relevant question and perhaps the only currently available dataset to answer this question. This is obviously relevant for management-related strategies.

The reviewer raises a valid issue, so we tested whether the result of our classification changes with different windows of observation. The results are shown in Figure R1. In Figure R1(A), we performed our analysis for variable time windows starting in 2009, including the entire 13-year study period (2009-2021), 12-year period (2009-2020), ..., and 8-year period (2009-2016), which is the mean period in Jankowski et al., 2017 (~8 years). We did the same analysis for periods ending in 2021 (Figure R1(B)), including the entire study period, 12-year period (2010-2021), ..., and 8-year period (2014-2021). We then examined the percentage of “unsafe” sites based on the same classification scheme shown in Figure 3 and the results are relatively stable, all showing ~80% “unsafe” sites.

Figure R1. The percentage of “unsafe” sites with different study period from 2009 (A) and end at 2021 (B).

Along these lines, it would also be interesting to see shorter time scale wetland surface elevation gain rates and their variability.

In their supplementary materials, Jankowski et al. (2017) tested how the rate of wetland surface elevation change varies with the time scale. They concluded that at least 5 years of surface elevation change data are needed to get a robust rate estimation. We have added this 5-year threshold in the main text in lines 141-142, which now reads as follows:

“Our SEC time series average 12 years in length, i.e., well over the five years previously found to be the minimum to obtain robust SEC data¹⁹.”

Minor comments:

Figure 3: It would be helpful to see these CRMS station examples identified somewhere on a map. This would also help visualize the observed variations in tidal range and wetland surface elevation gains. I would also recommend adding more spaces in the text on the left cartoons, as it seems a bit close together.

We modified and updated Figure 3 as the reviewer suggested and added Extended Figure 6 to show the location of the CRMS sites illustrated in Figure 3.

Figure 4A: I recommend changing the color scheme for the give-up scenarios, perhaps by adding a symbol in the dot. They are a bit hard to differentiate as is.

We thank the reviewer for this comment. However, we prefer to keep figure 4A in its original format, which passed the colorblind check.

Figure 4C: Can error bars be included with these data points?

Revised as the reviewer suggested.

Figure 5B: I recommend putting each IPCC scenario label also in each panel box for B-D
Added as the reviewer suggested.

1. We added a sentence in line 142-144 as shown below in bold italic format. We also use cRWL hereinafter for RWL with VLM correction to avoid confusion.

“In addition, we corrected the RWL data by removing the vertical land motion occurs between the bottom of the water-level gauge and RSET (see Methods).”

2. We updated the median deficit and 25th percentile deficit in Figure 5, and the values in line 226, because we re-ran the Monte Carlo simulation during the revision.
3. We revised the Figure 1.

REVIEWERS' COMMENTS

Reviewer #1 (Remarks to the Author):

I have re-read the manuscript by Li et al. and their response to the reviews. I supported publication in my original review but was concerned that the manuscript did not sufficiently recognize the similarities in their results and those from other studies. The authors have dealt with this concern and my other concerns. I note that they have also addressed concerns from the other reviews. This being the case, I recommend publication.

Reviewer #2 (Remarks to the Author):

Overall the paper is well written and based on a very impressive dataset. The natural experiment is a great set-up. The authors have satisfied all my previous comments except for the following minor suggestions...

A couple minor comments:

- 1.) The new first paragraph needs references supporting each of those stated impacts
- 2.) Line 168 - but isn't something else driving the changes in the Chenier Plane (see your line 133)?
- 3.) Line 187 - what is the topographic barrier in the case of the Mississippi Delta?
- 4.) 209 - Please give the number (n= X) for those with considerable organic components like you did for the other "outliers" (e.g. x = 12)

Reviewer #3 (Remarks to the Author):

The authors have addressed all of my initial review comments.

Reviewers' original comments are included below in black text

Manuscript Authors' responses to Reviewers' comments are included below the relevant comment in blue, bold text.

Reviewer #1 (Remarks to the Author):

I have re-read the manuscript by Li et al. and their response to the reviews. I supported publication in my original review but was concerned that the manuscript did not sufficiently recognize the similarities in their results and those from other studies. The authors have dealt with this concern and my other concerns. I note that they have also addressed concerns from the other reviews. This being the case, I recommend publication.

We thank the reviewer for the encouraging words!

Reviewer #2 (Remarks to the Author):

Overall the paper is well written and based on a very impressive dataset. The natural experiment is a great set-up. The authors have satisfied all my previous comments except for the following minor suggestions...

We thank the reviewer for their critical review and positive feedback on our manuscript!

A couple minor comments:

1.) The new first paragraph needs references supporting each of those stated impacts

Done

2.) Line 168 - but isn't something else driving the changes in the Chenier Plane (see your line 133)?

Yes, as we stated in line 121, this area is heavily impacted by human activities as well.

3.) Line 187 - what is the topographic barrier in the case of the Mississippi Delta?

In Figure 4A, we show the elevation data in our base map; the topographic barrier is the boundary where the Pleistocene surface is exposed, and the slope is relatively steep. Only in the relatively narrow Lower Mississippi Valley, this feature is lacking (i.e., the slope is gentler).

4.) 209 - Please give the number (n= X) for those with considerable organic components like you did for the other "outliers" (e.g. x = 12)

Done

Reviewer #3 (Remarks to the Author):

The authors have addressed all of my initial review comments.

We thank the reviewer for their critical review and positive feedback on our manuscript!